# CAFA: Coding as Auto-Formulation Can Boost Large Language Models in Solving Linear Programming Problem

**Haoxuan Deng, Bohao Zheng, Yirui Jiang, Trung Hieu Tran**
Cranfield University, United Kindom
{haoxuan.deng, bohao.zheng, yirui.jiang, t.h.tran}@cranfield.ac.uk

## Abstract

Large language models (LLMs) open new doors for Operations Research (OR). While initial studies explored multi-agent strategies for LLMs in OR, our research challenges the assumption that such complex multi-step pipelines unnecessarily yield superior results for Linear Programming (LP) problems. This paper introduces a streamlined methodology: Coding as Auto-Formulation (CAFA). In comparison, CAFA is only one compact prompt guiding the LLMs to formalize the given problem text into lines of codes. The generated code will be post-processing for execution to get the answer. The proposed methods is tested on the NL4OPT dataset with different LLMs. Results suggest that despite its simplicity, consistently enhances LP problem-solving accuracy across different models. This study aims to shed light on better unleashing LLMs' mathematical reasoning capability with more streamlined prompts. The code of this paper can be found in `https://github.com/BlueAsuka/CAFA`

## 1 Introduction

Large Language Models (LLMs) have transformed natural language processing with their ability to understand, analyze, and generate human-like text. Their success has led to increasing interest in applying LLMs to solve mathematical word problems Cobbe et al. [2021], drawing attention from the Operations Research (OR) field. Traditionally, solving OR problems like linear programming (LP) involves substantial human expertise to extract and formalize information for establishing optimization models and to use commercial solvers (CPLEX, Gurobi, etc.) to get the solution. The rise of LLMs offers the potential to automate this process, leading to the development of a new research area: LLMs for operations research (LLM4OR). Given that the research on LLM4OR is still in its early stages, this paper will narrow its focus specifically to the LP problem. Mathematically, LP is maximizing or minimizing a linear objective function, subject to linear inequality and equality constraints. It can be expressed as:

$$\min \text{ or } \max \ \mathbf{c}^T\mathbf{x}, \ \text{ where } \mathbf{A}\mathbf{x} \le \mathbf{b}, \ \mathbf{x} \ge 0 \tag{1}$$

Where $\mathbf{x}$ is the vector of decision variables, $\mathbf{c}$ represents the coefficients of the objective function, and $\mathbf{A}$ and $\mathbf{b}$ define the parameters and limits of constraints.

Recent research on this topic has explored various methods. The main idea is to carefully design instructions to guide the LLM to decompose the optimization process into a series of sub-tasks and solve them sequentially to derive the answer. Most of the research follows the abstract workflow illustrated in Figure 1. Typically, the workflow mainly includes four components: 1). An analyzer for name-entity recognition of variables and relations in the given text for objectives and constraints

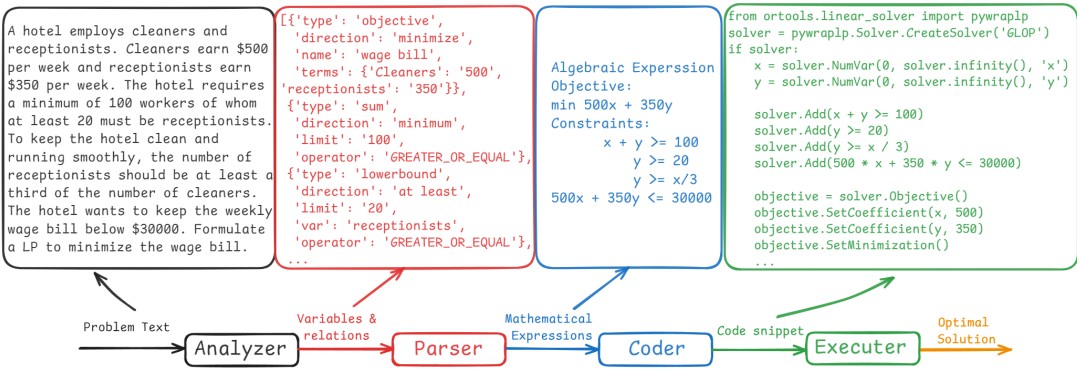

Figure 1: The abstract diagram of LLMs for the linear programming optimization automatic workflow

extraction. 2). A parser to formulate the extracted information into mathematical expressions. 3). A coder to compose a code snippet to call external solvers for problem-solving. 4). An executor (usually using a Python interpreter) runs the code to obtain the final solution.

Remarkable works include Chain-of-Expert (CoE) Xiao et al. [2023] and OptiMUS AhmadiTeshnizi et al. [2024]. Both of these approaches employ multi-agent systems based on LLM by assigning multiple expert agents with distinct roles to implement the mentioned components of the LP optimization process to get the solution. OptiGuide Li et al. [2023] extends the applications of the proposed method to supply chain optimization, allowing non-technical users in the field of logistic to use optimization packages.

Although the effectiveness of these methods have been demonstrated, their implementations rely on autonomous agent workflows that demand extensive prompt engineering and cutting-edge models like GPT-4 or Claude2. This raises two key questions: (1) Is a multi-agent framework with intensive prompt engineering essential for unlocking LLMs' problem-solving potential in LP tasks? (2) How can less-capable LLMs (open source models) be leveraged to reason and solve LP problems with reduced prompt engineering requirements? This paper seeks to address these two challenges.

The remainder of this paper is organized as follows. Section 2 presents a probabilistic framework to model the multi-agent water flow diagram used for LP automation and analysis. Section 3 provides an in-depth discussion of the motivation behind and the design details of the CAFA framework. In Section 4, experimental evaluations are conducted on the NL4Opt dataset, with performance comparisons against selected baseline models. Section 5 addresses the limitations of the current approach and discusses potential directions for future research. Finally, the paper concludes with a brief summary of key findings.

## 2 Problem Formulation and Analysis

Consider a Linear Programming (LP) problem presented in text, denoted as $Q$, and a language model $L_\theta$ tasked with generating the correct answer $A$, where $\theta$ represents the fixed parameters of the pretrained model. The probability of obtaining the correct answer $A$, given the question $Q$, is expressed as $P_\theta(A|Q)$. For simplicity, when using a single model $L_\theta$ in a multi-step process, this probability can be abbreviated as $P(A|Q)$.

In a multi-agent framework, the final answer $A$ is produced over $n$ steps, resulting in $A = A_n$. At each step $i$, an intermediate result $A_i$ is generated based on the corresponding prompt $p_i$. Applying the chain rule, the probability of the final answer is decomposed as follows:

$$P(A|Q) = P(A_n|Q) = \prod_{i=2}^{n} P(A_i|A_{i-1}, p_i)P(A_1|Q, p_1) \qquad (2)$$

In a multi-agent system, it is often the case that the prompt $p_i$ is explicitly predefined and remains unchanged regardless of the answers generated in previous steps. Consequently, it is reasonable

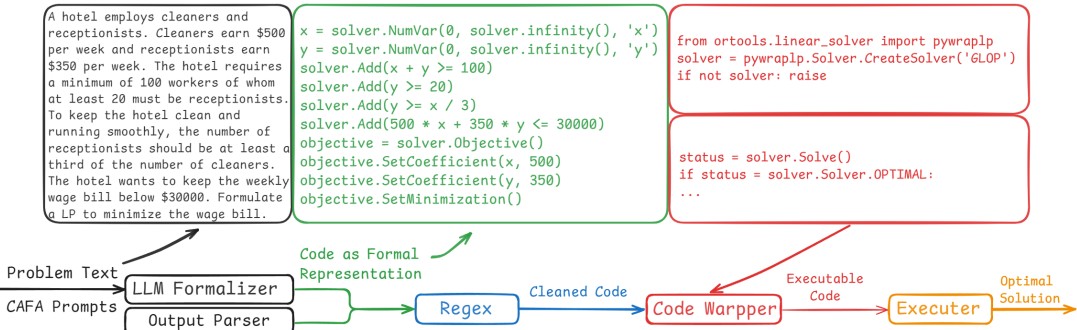

Figure 2: LP solving with the CAFA prompt and external code post-processing

to assume that $p_i$ is independent of prior answers $A_{i-1}, A_{i-2}, ..., A_1$. At each step, the answer $A_i$ depends solely on the current prompt $p_i$, eliminating the need to track the entire history of previous answers. Thus, the probability expression simplifies as $P(A_i|A_{i-1}, p_i) = P(A_i|p_i)$, allowing for a more streamlined version of the equation.

$$P(A|Q) = P(A_1|Q, p_1) \prod_{i=2}^{n} P(A_i|p_i) \tag{3}$$

Since $\prod_{i=2}^{n} P(A_i|p_i) \leq 1$

$$P(A|Q) \leq P(A_1|Q, p_1) \tag{4}$$

In summary, the initial prompt $p_1$ is crucial in determining the correctness of the final answer. Moreover, the multi-step workflow does not always enhance performance, as subsequent steps may introduce error accumulation, which can degrade the overall accuracy of the solution. This thus answer the first question: chunking the LP solving procedure into multiple sequential steps and solve them one-by-one is not always lead to a better performance in the LP task.

It is a note that equation 4 is not conflict to the main idea of Chain-of-Through (CoT), since equation 4 demonstrates the contribution of various prompts across sequential tasks, while CoT is only one prompt for a single task.

## 3 Code as Auto-Formulation Prompting with Code Post-Processing

According to the equation 4, the key lies in shrinking the problem-solving process into one step with well-design first prompt to obtain a better performance. For the LP problem-solving, it is unrealistic to enable the LLM to obtain correct answer directly by just zero-shot prompting. An alternative way is to let the LLM to generate an intermediate presentation (IR) of the text to disambiguate the problem for processing. According to Ramamonjison et al. [2022], they use two-stages approach to convert the text into a context-free expression, specifically, a matrix representation of the constraints and objectives reported in the text. Then, the matrix is used to parse code to call solver to obtain the answer. Notice that the code is the expected output from the LLMs in the final step in all mentioned framework, to reduce the number of reasoning steps, we can instruct the LLM to generate the code in the one step as the formal representation. This idea is similar to Program-of-Through (PoT) Chen et al. [2023], which also use code during reasoning instead of text. In our work, we want the LLMs to only generate code that can capture the constraints, parameters, limits, and objectives mentioned in the text.

The complete LP-solving process is illustrated in Figure 2. Initially, the problem text, along with the CAFA prompt, is input into the LLM for code generation. Additionally, a Pydantic parser is employed to ensure that the LLM generates code adhering to the specified format, thus enhancing consistency and reducing the likelihood of extraneous or incorrect string generation. The generated code represents a formal translation of the textual problem using the Python syntax of the Guribo

Table 1: Accuracy of different methods with various LLMs on the NL4Opt dataset

|  | Standard | Chain-of-Expert | OptiMUS | CAFA |
|---|---|---|---|---|
| GPT-4 | 47.3% * | 64.2% * | **78.7%** * | 70.1% |
| GPT-3.5-Turbo | 42.4% * | 58.9% * | 28.6% * | **59.0%** |
| DeepSeek-Coder v2 | 5.2% | - | - | **60.1%** |
| Llama 3.1 | 5.5% | - | - | **34.0%** |

* referred to the results reported in the original paper.

solver; however, this code is not executable at this stage. In the subsequent step, a regular expression (regex) is applied to validate the syntax and correct any errors by refining unwanted patterns. The revised code is then encapsulated with the necessary suffixes and prefixes, transforming it into executable code. Finally, this code is executed via the Python interpreter to derive the final result (details regarding the CAFA prompt and regex rules are provided in the Appendix).

## 4 Experiment and Results

To evaluate the proposed methods, we utilize NL4Opt dataset originally introduced by Ramamonjison et al. [2022]. The baseline methods selected for comparison include Chain-of-Expert and OptiMUS. To assess the generalization of the approach across different models, we conduct experiments using **GPT-4**, **GPT-3.5-turbo**, **deepseek-coder-v2-16B**, and **Llama3.1-8B**.

For performance evaluation, we employ **Accuracy** as the primary metric. If the output matches the correct answer provided in the dataset, the result is marked as correct for that specific question. Otherwise, it is considered incorrect. Accuracy is calculated as the ratio of correct solutions to the total number of questions.

Table 1 demonstrate that the proposed method enhances performance in solving LP problems across various models. For GPT-3.5-Turbo and GPT-4, the CAFA method enables them to achieve competitive results compared to state-of-the-art approaches. Specifically, GPT-3.5-Turbo achieves 59% accuracy, outperforming both the Chain-of-Expert (CoE) and OptiMUS methods when using the same model. GPT-4 attains 70.1% accuracy, surpassing the CoE method, though lower than OptiMUS. However, unlike OptiMUS, CAFA requires only a single prompt, making system management, tuning, and optimization more straightforward for further improvement.

For smaller LLMs, such as DeepSeek-Coder 16B and Llama 3.1 8B, the CAFA method significantly improves performance. Notably, DeepSeek-Coder achieves a performance level similar to GPT-3.5-Turbo, with 60% accuracy. This suggests that LLMs with strong coding capabilities can yield better results, as the formal translation of problem text into executable code is closely tied to the model's coding proficiency. This step is crucial in the proposed method.

Based on this idea and Equation 4, the capability to solve mathematical problems may be strongly correlated with the model's ability to generate code for formal problem representation. Future research could explore how to further leverage LLMs' coding capabilities to enhance performance. Additionally, investigating alternative representations beyond code formatting, such as relation triples, graphs or symbolic equation, could offer insights into improving the quality of auto-formulation and achieving higher accuracy.

## 5 Limitations and Future Works

**Iterative Correction Mechanisms.** Based on the experimental results conducted on the NL4Opt dataset, while the CAFA framework demonstrates competitive performance, it falls short of achieving the state-of-the-art results demonstrated by GPT-4. This discrepancy can be mainly attributed to the simplified independent assumption used to derive the equations referenced in Equation 3. As widely observed in multi-agent systems, problem-solving processes do not follow a straightforward, one-pass pipeline but rather involve more complex, iterative network architectures. Such systems allow for iterative tracking, refinement, and correction of errors throughout the problem-solving process, leading to an information dependence across multiple steps. A promising direction for future research

would involve the introduction of iterative correction mechanisms within the intermediate code representations of problems, potentially offering improved performance over one-pass translation approaches.

**Automatic Prompt Engineering and Exemplars Selection.** Handcrafting instructions within the CAFA prompt remains necessary to achieve satisfactory results. This presents a limitation to the generalizability of the CAFA when applied to other types of optimization problems, such as mixed integer linear programming (MILP) or quadratic programming (QP). A key challenge is how to enable automatic prompt engineering (APE) to accommodate various optimization problems with minimal human intervention. In addition, identifying optimal examples used as few-shot demonstrations in the prompt represents another critical area of exploration. While there has been groundbreaking research in APE for arithmetic problems Yang et al. [2023], Khattab et al. [2023], investigations into its application in the domain of operations research (OR) remain scarce. This gap highlights an important opportunity for future research direction.

**Various Optimization Problems and Multi-modality Support.** The CAFA prompt presented in this paper is designed specifically for linear programming problems with no more than three variables. This is too simplified and limits its applicability to many real-world scenarios. To meet the diverse demands and objectives of various applications, it is necessary to extend the capabilities of the proposed CAFA prompt. An initial extension involves adapting it to handle other types of optimization problems, such as mixed integer linear programming (MILP) and quadratic programming (QP). This also call for curating datasets of multiple types of OR problems. Furthermore, the current CAFA prompt processes only text input, whereas future research could explore a multimodal CAFA capable of supporting diverse or mixed input formats, including text, images, and tabular data. Such advancements would enable CAFA to address a broader range of needs across various use cases.

## 6    Conclusion

This paper reviews several LLM4OR approaches that utilize a multi-agent framework with large language models (LLMs). An analysis of multi-step water flow pipelines using probabilistic modeling highlights the pivotal role of the initial prompt in achieving accurate solutions. It underscores that increasing the number of steps can negatively impact final performance in Linear Programming (LP) tasks. Based on this analysis, we introduce the CAFA (Code-as-Auto-Formulation) prompting method, which formalizes problem text into executable code in a single step. With code post-processing, various LLMs are able to improve their performance in LP problem-solving. Experimental results indicate that even lower-capacity open-source models benefit from the proposed method when paired with simplified prompt engineering. **By CAFA, LLMs' mathematical problem-solving ability is closely linked to its capability for text-to-code formalization.** Future research directions include automating prompts and exemplar selection with minimal human intervention. Additionally, exploring alternative formats for formal representations (relation triplet, graph, symbolic equation, etc.) could potentially improve the performance of LLMs in problem-solving tasks.

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

# A CAFA Prompt

```
You are an expert in optimization problems and domain specific
    language generation.
Your task is to convert the textual optimization text into lines of
    code.
You should also analyze whether the variable in the optimization
    problem should be INTEGER or CONTINUOUS.
DO NOT ADD ANY COMMENTS OR EXPLANATION TO THE CODE. JUST OUTPUT THE
    CODE.
Here are some examples that you should refer to:

QUESTION:
A car manufacturer makes two types of car oils: Oil Max and Oil Max
    Pro. A container of Oil Max contains 46 grams of substance A, 43
    grams of substance B and 56 grams of substance C. A container of
    Oil Max Pro contains 13 grams of substance A, 4 grams of substance
     B and 45 grams of substance C. The car manufacturer has 1345
    grams of substance A, 346 grams of substance B, 1643 grams of
    substance C. In addition, the profit per container of Oil Max is
    $10 and the profit per container of Oil Max Pro is $15. How many
    containers of each of oil should the car manufacturer make to
    maximize profit?
CODE:
x = m.addVar(name="Oil Max", vtype=gp.GRB.INTEGER)
y = m.addVar(name="Oil Max Pro", vtype=gp.GRB.INTEGER)
m.setObjective(10 * x + 15 * y, gp.GRB.MAXIMIZE)
m.addConstr(46 * x + 13 * y <= 1345)
m.addConstr(43 * x + 4 * y <= 346)
m.addConstr(56 * x + 45 * y <= 1643)

QUESTION:
Ben is growing apples and pears on his orchard. He has 50 acres
    available on which he must grow a minimum of 5 acres of apples and
     a minimum of 10 acres of pears to meet demands. The profit per
    apple is $2 and the profit per pear is $4. He prefers to grow more
     pears than apples but limitations in his workforce allow him to
    grow at most twice the amount of pears as apples. How many of each
     fruit should Ben grow in order to maximize his profit? What is
    that profit?
CODE:
x = m.addVar(name="apples", vtype=gp.GRB.INTEGER)
y = m.addVar(name="pears", vtype=gp.GRB.INTEGER)
m.setObjective(2 * x + 4 * y, gp.GRB.MAXIMIZE)
m.addConstr(x + y <= 50)
m.addConstr(x >= 5)
m.addConstr(y >= 10)
m.addConstr(y <= 2 * x)

Please finish the task think step by step.
QUESTION:{q}
```

# B Code Post-Processing

The code post-processing includes using regular expression to check, filter and correct the generated code from the LLM, simplfied inequalities in the code, and complement to piece of for execution. The following code snippet is the regex function.

```
{\tin
def clean_code(code: str) -> str:
    temp_code = code
    # Split the code into lines
    pattern = r'\)([a-zA-Z])'
```

```python
        temp_code = re.sub(pattern, r')\n\1', temp_code)

    cleand_code = []
    for line in temp_code.split('\n'):
        line = line.strip()
        # Replace > < to >= <=
        if line.startswith('m.addConstr') and not re.findall(r'<=|>=',
 line):
            # print("Not found")
            line = re.sub(r'<', r'<=', line)
            line = re.sub(r'>', r'>=', line)
        # Remove all comments and suffix and prefix terms
        if not line.startswith('```') and not line.startswith('#'):
            cleand_code.append(line)
        else:
            continue
        # Don't support bool expression '=='
        if re.findall(r'==', line):
            cleand_code.remove(line)

    cleand_code = '\n'.join(cleand_code)

    # Remove all '{' and '}'
    cleand_code = cleand_code.replace('{', '').replace('}', '')
    return cleand_code
}
```

The following function is used to simplify ineqaulity

```python
def simplify_code(code: str) -> str:
    simplfied_code = []
    for i, line in enumerate(code.split('\n')):
        if line.startswith('m.addConstr') or line.startswith('m.
    setObjective'):
            if '/' in line:
                obj_pattern = r'm\.setObjective\(([^,]*)'
                constr_pattern = r'm\.addConstr\((.*)\)'
                if re.findall(obj_pattern, line):
                    matches = re.findall(obj_pattern, line)
                    obj = re.search(r'gp\.GRB\.(\w+)', line).group(1)
                    expr = sp.sympify(matches[0])
                    simplfied_code.append(f"m.setObjective({str(sp.
    simplify(expr))}, gp.GRB.{obj})")
                if re.findall(constr_pattern, line):
                    matches = re.findall(constr_pattern, line)
                    oper = re.search(r'\s*(>=|<=)\s*', matches[0]).
    group(1)
                    expr = sp.sympify(matches[0])
                    simplified_expr = str(sp.simplify(expr.lhs - expr.
    rhs))
                    if match := re.search(r'^\((.*?)\)/',
    simplified_expr):
                        new_constr = f'{match.group(1)} {oper} {str(0)
    }'
                        simplfied_code.append('m.addConstr(' +
    new_constr + ')')
            else:
                simplfied_code.append(line)
        else:
            simplfied_code.append(line)
    return '\n'.join(simplfied_code)
```

The following functions is used to complment and execute the code

```python
prefix = """
```

```python
import gurobipy as gp
env = gp.Env(empty=True)
env.setParam("OutputFlag",0)
env.start()
m = gp.Model(env=env)
"""

suffix = """
m.optimize()
"""

def complement_code(code: str) -> float:
    return prefix + code + suffix

def execute_code(code: str) -> float:
    ex_locals = {}
    exec(code, None, ex_locals)

    try:
        return ex_locals["m"].objVal
    except Exception as e:
        return np.inf
```

The final answer can be run by the following code

```python
ans = execute_code(
        complement_code(simplify_code(clean_code(code_str)))
     )
```

