# OpenReview forum: "CAFA: Coding as Auto-Formulation Can Boost Large Language Models in Solving Linear Programming Problem"
_NeurIPS.cc/2024/Workshop/MATH-AI — MATH-AI 24_

### Official Review · Reviewer_MNoe · 2024-09-28
**A Simplified Approach for LP Problem-Solving with LLMs**

**Rating:** 5
**Confidence:** 4

**Review:**

## Summary

This study introduces CAFA (Coding as Auto-Formulation), a streamlined approach for solving Linear Programming (LP) problems using Large Language Models (LLMs). CAFA uses a single prompt to guide LLMs in formalizing problem text into executable code, followed by code post-processing and execution to obtain the solution.

## Pros

1. CAFA simplifies the LP problem-solving process by reducing it to a single-step formalization, achieving competitive or superior performance compared to more complex multi-agent approaches across various LLMs, including smaller open-source models.

## Cons

1. While CAFA demonstrates improvements across multiple models, it still falls short of the best-performing multi-agent method (OptiMUS) when using GPT-4.
2. The paper primarily focuses on linear programming problems, leaving the applicability and effectiveness for more complex optimization problem types (such as non-linear programming, integer programming, etc.) unexplored.
3. The study lacks comparison with the latest software agents, such as SWE-Agent and Data Interpreter.

---

### Official Review · Reviewer_dimq · 2024-10-03
**Review of paper 37 - CAFA: Coding as Auto-Formulation Can Boost Large Language Models in Solving Linear Programming Problem**

**Rating:** 7
**Confidence:** 4

**Review:**

**Summary of contributions:**

This paper brings in Coding as Auto-Formulation (CAFA) with the objective of enhancing LLMs’ mathematical cognitive capability with more streamlined prompts. The study focusses on the fact that instead of complex multi-step pipelines, CAFA allows LLMs to formalize problem text into lines of code through a single concise prompt.

**Relevance and Clarity:**

Relevant to the guiding theme of the proposed workshop as the authors explore to add new capabilities to the LLMs used for Operations Research.

**Originality:**

 The authors come up with a streamlined methodology to transform error prone multi-step strategies to a single step framework to tackle Linear Programming problems and report performances which are comparable or even better than that of SotA techniques like Chain-of-Expert and OptiMUS.

**Strengths**

1. *Research Questions:* The research questions are well framed and should succeed to generate interest among the audience. Particularly, the second research question “How can less-capable LLMs be leveraged to reason and solve LP problems with reduced prompt engineering requirements?” appears appealing as many open source LLMs have actually opened up exciting possibilities for solving Linear Programming problems in Operations Research.


2. *Problem mitigation:* Problem formulation and mitigation are well explained through diagrams and equations. Instead of using a two-stage approach, the idea to enable the LLMs to generate code in one step can go a long way in improving performance of LLMs in Operations Research.

 **Weakness**

1. The accuracies reported by the authors are on NL4Opt dataset. The authors should also test their method on multiple diverse datasets to prove its generalizability.

2. Sentence construction and grammatical correctness have avenues for improvement. This is important as it will help to convey precise meaning and engage the readers, leaving a positive impression.

**Significance**

The authors report significant improvement in accuracy for the smaller LLMs through CAFA and hence this research can play a significant role in context with limited resources for Operations Research like using open source LLMs.

---

### Official Review · Reviewer_XSXs · 2024-10-06
**This paper introduces CAFA (Coding as Auto-Formulation), a simplified approach for using large language models (LLMs) to solve linear programming (LP) problems.**

**Rating:** 7
**Confidence:** 4

**Review:**

The key contributions are:

A theoretical analysis showing that multi-step pipelines do not necessarily improve performance over single-step approaches for LP tasks.
A streamlined prompting method that guides LLMs to directly generate code representing the LP problem formulation in one step.
Experimental results demonstrating CAFA's effectiveness across different LLM sizes, including open-source models.

Pros:

The theoretical analysis provides valuable insights into the limitations of multi-step approaches.
CAFA achieves competitive performance with a much simpler prompting strategy compared to prior work.
The method generalizes well to smaller open-source models, expanding potential applications.
The approach of using code generation as an intermediate representation is novel and promising.

Cons:

The theoretical analysis, while insightful, makes some simplifying assumptions that may not fully capture real-world scenarios.
Performance on GPT-4 is lower than the state-of-the-art OptiMUS method, though with less complexity.
Limited exploration of alternative intermediate representations beyond code.
The paper could benefit from more in-depth error analysis and discussion of limitations.

The paper is clearly written and well-structured. The motivation and approach are described logically, and the experimental results are presented clearly. The work makes a meaningful contribution to the growing field of using LLMs for operations research tasks.
The CAFA method's simplicity and generalizability are particularly noteworthy. By reducing the problem to a single code generation step, it avoids error accumulation issues inherent in multi-step approaches. The strong performance on smaller models like DeepSeek-Coder is especially promising for practical applications.

However, there are some limitations to consider. The theoretical analysis, while providing useful insights, relies on some simplifying assumptions. A more nuanced discussion of when these assumptions may break down would strengthen the paper. Additionally, while CAFA performs well, it does not surpass state-of-the-art performance on GPT-4. A more detailed comparison and discussion of trade-offs with methods like OptiMUS would be valuable.

The paper opens up interesting directions for future work, particularly in exploring alternative intermediate representations and further leveraging LLMs' coding capabilities for mathematical reasoning tasks.

Overall, this is a solid contribution that introduces a simple yet effective approach for LP problem-solving with LLMs. The method's generalizability and the insights provided by the theoretical analysis make it a valuable addition to the literature.

---

### Decision · Program_Chairs · 2024-10-08

Accept